# The Pathobiological Underpinnings of Psychosis: From the Stress-Related Hypothesis to a Multisystemic Approach

**DOI:** 10.3390/neurosci6040099

**Published:** 2025-10-03

**Authors:** Evangelos Karanikas

**Affiliations:** Department of Psychiatry, 424 General Military Hospital, 564 29 Thessaloniki, Greece; e.p.karanikas@army.gr

**Keywords:** immune, inflammation, metabolism, neuroendocrine, psychosis, redox

## Abstract

Until recently, research on the pathobiological substrate of psychosis has been focused on neurotransmitter perturbations. However, this scope has expanded to include new fields, such as the immune/redox/metabolic/neuroendocrine/stress systems. Indeed, basic research in the stress field showed that the systems above can represent components of a general inflammatory process as tightly interconnected as a Gordian knot. Based on the inflammatory hypothesis concerning the psychosis etiopathology, the findings from psychotic cohort studies on each one of the immune/redox/metabolic/neuroendocrine/stress systems have started to accumulate. The evidence favors the involvement of these systems in the formation of the pathobiological psychotic substrate, yet little is known concerning their interplay. This review attempts to establish a frame of reference for the evidence concerning intersystemic interactions, starting with the basic research on the stress field and expanding to clinical studies with psychosis cohorts, hoping to instigate new avenues of research.

## 1. Introduction

Psychosis constitutes one of the most debilitating conditions of human health pathology, with an estimated 4.6 and 7.49 per 1000 persons pooled median global prevalence and median lifetime prevalence, respectively [1]. Psychotic symptomatology presents at different levels and dimensions (positive, negative, cognitive). This is similar to the underlying pathobiology initially used to incorporate hypotheses in terms of neurotransmitter imbalance, namely the excessive stimulation of the presynaptic Dopamine2-Receptors (D2-Rs) in the striatum [2], Glutamate, and NMDA-R hypofunction [3] and the perturbation of GABA-producing Parvalbumin Interneurons (PVIs) impacting glutamatergic neurons and subsequently dopaminergic ones [4], as well as other neurotransmitter systems, such as 5-HT, kynurenines, Acetylcholine, and adenosine [4,5,6]. Lately, research on the psychosis pathobiology has expanded to additional fields, such as the immune, redox, metabolic, and neuroendocrine/stress systems [7,8]. Despite the relative abundance of preclinical findings advocating a close relationship of stress with immune, redox, metabolic, and neuroendocrine/stress components and their in-between complex, as well as tight interactions [9,10,11,12,13,14], only lately has the focus of research been directed towards the involvement of each one of the immune, redox, metabolic, and neuroendocrine/stress systems in psychosis [8,15,16,17,18,19]. The hitherto prevailing practice has been to research each one of these systems separately, hence the gap in the literature concerning their interactions in psychosis. Again, the reader needs to be mindful that any attempt to segregate immune/redox/metabolic/neuroendocrine/stress scopes is conventional and schematic, meant to serve the practicality of the review and the comprehension of underlying mechanisms. The hypothesis underlying this review is that the various scopes discussed are closely interconnected and exhibit different directions of causality. It is proposed that the components within each field not only relate to other components in the same field at different levels—through feedforward, mediating/moderating, or bottom-up/top-down processes—but also interact with components from other fields under review in a similarly multi-directional manner. This review aims to explore the assumption that these biological fields can modify their structural and functional states under chronic or intense stress, leading to alterations in pathobiological pathways. These changes may impact and be influenced by brain structures, thereby linking them to the pathobiology of psychosis. In reality, these fields appear closely interdependent, representing various snapshots of the same general inflammatory process [20]. To accommodate the needs of this narrative review with an exploratory character, the author aggregated evidence from preclinical studies on the immune/redox/metabolic/neuroendocrine/stress systems’ involvement in brain function alterations in the context of stress and subsequently expanded with clinical studies on cohorts with psychosis, with an evaluation of parameters from the immune/redox/metabolic/neuroendocrine/stress systems. Particular attention was given to any evidence suggesting intersystemic interactions.

## 2. Immunity

The immune system orchestrates the organism’s survival against physiological and psychological stress, exerting its effects on multiple systems and the brain itself through an impaired Blood–Brain Barrier (BBB) [14,21]. Pathogen/Damage-Associated Molecular Patterns (PAMPs, DAMPs) prime microglia, the main immune effectors within the brain parenchyma, via their action on Toll-Like Receptors (TLRs). This leads to the activation of the nuclear factor kappa-light-chain-enhancer of activated B cells (NF-κB), Activator Protein-1 (AP-1), the nucleotide-binding domain, the leucine-rich-containing family, and pyrin domain-containing-3 (NLRP-3) inflammasome. The next stage involves the activation of Mitogen-Activated Protein Kinases (MAPKs), followed by that of microglia producing pro-inflammatory cytokines within (but not limited to) the brain [10,11,22].

Inflammatory compounds in the brain appear to disrupt neurotransmission, as evidenced by the Maternal Immune Activation (MIA) model showing the post-viral-induced penetration of maternal pro-inflammatory cytokines into the fetal brain [23]. Indeed, in offspring, the MIA paradigm produces a symptomatology reminiscent of psychosis and reveals neurotransmitter aberrations such as hippocampal NMDA-R hypofunction, enhanced Dopamine turnover, and GABA interneuron migration disruption, all furthering MIA’s translational potential [24,25,26,27].

Regarding the involvement of immunity in the formation of the pathobiological substrate in psychosis explored via clinical studies, initial evidence suggested a differential cytokine signature associated with the clinical status. Specifically, Miller et al., in 2011, [18] suggested that IL-1, IL-6, and TGF-β could serve as potential state markers, while IL-12, IFN-γ, TNF-α, and sIL-2R were identified as trait markers. Subsequent meta-analyses revealed that a pro-inflammatory immune signature—characterized by IL-1β, sIL-2R, IL-6, TNF-α, IFN-γ, IL-6, IL-12, and IL-17—distinguishes the First Episode Psychosis (FEP) stage, even prior to the initiation of medication [28,29]. Additionally other studies supported the hypothesis that there is a synchronous activation of all arms of the immune system including anti-inflammatory responses [17,30,31], which was extensively reviewed by Roomruangwrong et al. (2019) [32].

Evidence of the role of immunity in the formation of the pathobiological substrate in psychosis has appeared even before the FEP stage emerges. Recent studies on Ultra-High Risk for Psychosis (UHR-P) cohorts demonstrated increased peripheral inflammation, although they presented with varied cytokine signatures, (IFN-γ, IL-10, IP-10/CXCL10, MCP-1/CCL2, MIP-1β/CCL4, Eotaxin/CCL11, TARC/CCL17, MDC/CCL22) [33], (IL-12/IL-23p40) [34], and (IL-1β, IL-7, IL-8) [35].

Moreover, research findings have attributed predictive capacity, relative to the transition to psychosis, to immune parameters.

Specifically, population-based studies from the UK, Finland, and Sweden attributed a predictive potential relative to psychosis emergence later in the lifespan to IL-6, CRP, and ESR [36,37,38]. As a supplement to the aforementioned work, Osimo et al. (2021) posited increased (>3 mg/L) CRP levels as an additional risk factor for psychosis [39].

The evidence from the medication field is also supportive of antipsychotics’ immunoregulatory role, which is compatible with the inflammatory pathobiological hypothesis in psychosis. Typical and atypical antipsychotics showed anti-inflammatory potential, as evidenced by in vitro immune-challenged human peripheral blood mononuclear cell cultures [40,41,42,43]. Clozapine, as well as second-generation antipsychotics such as olanzapine and risperidone, and lithium have exhibited downregulatory potential over NF-kB yet not consistently [44,45,46]. Thus, the pharmacological field provides evidence of the antipsychotics’ anti-inflammatory effect, hence corroborating the inflammatory etiopathological basis in psychosis. Interestingly, even antidepressant medications—through their action on the STAT/JAK pathway, which is thought to participate in the pathobiology of psychosis—may have an adjunctive therapeutic role against psychosis [47]. Beyond the conventional limits of psychotropic medications, other agents with immunomodulating properties, such as NSAIDs, COX inhibitors, minocycline, and N-Acetylcysteine (NAC), have shown beneficial effects in early psychosis as adjunct agents [48]. Gold-standard methods for refractory states in psychosis, such as Electroconvulsive Therapy (ECT), have only recently started being associated with immunomodulating effects, as evidenced by IL-6 and CRP decreases [49].

Lately, another marker of immune system dysregulation in various diseases, namely the Neutrophil-to-Lymphocyte Ratio (NLR), has been implemented in psychosis research [50]. Two meta-analyses showed increased NLR values both in the FEP and in multi-episode Schizophrenia (SCZ) [51,52], and a contemporary meta-analysis demonstrated increased Neutrophils in similar cohorts [53].

## 3. Redox

The brain’s energy requirement is vast, representing more than 20% of the total body’s consumption [54], and mitochondria produce the bulk of this energy via aerobic oxidative phosphorylation (OXPHO). The energy requirement increases under either stressful physiological or psychological circumstances. Accordingly, the redox process intensifies, culminating in Oxidative Stress (OXS), followed by a consequent generation of toxic by-products, namely the Reactive Oxygen/Nitrogen Species (ROS/RNS) [12,55]. When disinhibited due to insufficient anti-oxidants, the latter impose detrimental effects on DNA, lipids, and proteins [56], thus negatively impacting the neuronal membrane’s integrity [57,58]. There is substantial preclinical evidence favoring a close interaction between the redox system and neurotransmission yet with an equivocal directionality [13]. Indeed, constitutively speaking, baseline neuronal NO Synthase (nNOS) function along with sufficient NO and H_2_O_2_ concentrations are prerequisites for physiological NMDA-Rs augmentation [59,60,61]; however, their increased concentration, such as in the case of intense/prolonged stress, can perilously impact the NMDA-Rs of the Parvalbumin Interneurons (PVIs), culminating in dopaminergic signaling disruption [1,12,62]. In this hyperdopaminergic state, it is believed that ROS/RNS levels are further intensified due to the increased catabolism of Dopamine, creating a vicious cycle [63,64]. This condition can also result in hyper-glutamatergia and hypo-GABAergia, ultimately leading to cortical disinhibition [65,66,67].

The evidence leaning towards an increased OxS-generated toxic lipid and protein by-product burden, combined with impaired anti-oxidant defenses, suggests the involvement of the redox system in the formation of psychosis pathobiology [68]. Indeed, OXS-generated lipid peroxide (PX) by-products, such as Thiobarbituric Acid Reactive Substances (TBARSs) and Malondialdehyde (MDA), increased not only in chronic psychosis in both the brain and the periphery [69,70,71] but also in the early stage of the disorder [72,73,74,75].

As previously stated, apart from increased pro-oxidative processes, decreased anti-oxidant defense can constitute another pillar of OXS-related neurotoxicity in psychosis [69]. Indeed, anti-oxidant levels are aberrant in blood, Cerebrospinal Fluid (CSF), and post-mortem samples of psychosis cohorts [76,77]. Specifically, the Total Anti-Oxidant Status (TAS)—a cumulative index of all non-enzymatic endogenous as well as exogenous anti-oxidants, offering a comprehensive overview of the body’s oxidative defense status—decreased, even from the FEP stage in cross-sectional studies [73,78], meta-analyses, and reviews [15,16,71,76,78], yet not without evidence for the opposite effect [79].

Another pillar of the anti-oxidant defense system, Glutathione (GSH) and its oxidized form, GSH disulfide (GSSG), tended to be found to be decreased/increased, respectively, in post-mortem samples in the PFC and striatum [77,80], plasma [81], erythrocytes [82], CSF [83], and anterior cingulate cortex (ACC) in cohorts of subjects with established SCZ [84] as well as in the FEP [83,85,86,87,88].

Apart from the GSH system, there is an additional anti-oxidant enzymatic defense system, the thioredoxin (TRX) system, including the enzymes Superoxide Dismutase (SOD), Catalase (CAT), and GSH Peroxidase (GPx) [89]. Despite the prevalent notion of a GSH system compromise in psychosis, even from its early phase [8], the TRX system also appears to be perturbed [90]. This alteration (increase) in the TRX system function was attributed by the authors to high energy demands in leucocytes of FESCZ, based on findings indicative of the upregulation of all glycolytic pathways, from anaerobic glycolysis to the Krebs cycle and OXPHO.

The evidence favoring the dysregulation of the redox system in psychosis emerges even prior to the FEP stage. Indeed, cohorts of UHR-P subjects and unaffected siblings exhibited altered (decreased) activity of the anti-oxidant enzymes SOD and CAT both in serum and RBC, respectively, compared to healthy individuals [91,92]. Last but not least, the redox system’s aberration appeared to exert a predictive capacity concerning the transition to psychosis [35,93,94].

## 4. Metabolism

Metabolism refers to the whole sum of reactions taking place within each cell and aiming to provide the body with energy, thus accommodating vital processes, preservation, and new organic material synthesis. The brain’s vast energy requirements are covered principally by glucose, and its uptake accounts for more than 50% of the body’s glucose supply [95]. OXPHO mainly facilitates these requirements with oxygen in mitochondria, where the bulk of the cell’s energy is generated. Under stressful conditions, a malfunctioning Glyoxalase system can cause the glycolytic pathway to produce Advanced Glycation End-Products (AGEs). These compounds can further damage biomolecules [96]. Moreover, important biomolecules, such as neurotransmitters, can be negatively impacted by impaired glucose uptake in brain cells. This uptake is primarily regulated by Insulin signaling. A condition associated with decreased Insulin sensitivity is referred to as ‘type 3’ Diabetes [97]. This condition has also been linked to the inhibition of 5′ Adenosine Monophosphate-Activated Protein Kinase (AMPK). Given that AMPK is a cellular energy state sensor and a master regulator of the cell’s energy homeostasis—which is phosphorylated in conditions of low ATP bioavailability, such as hypoxia or acute neuronal activation [98,99], inducing Fatty Acid Oxidosis (FAO) and ketolysis [100]—it becomes evident that disturbances in Insulin sensitivity, i.e., Insulin Resistance (IR), can confer energy bioavailability aberrations at many levels, namely decreased glycolysis and ketogenesis as well as the synthesis of pro-apoptotic ceramides, thus resulting in the brain’s energy deprivation [101]. Besides central and systemic gluco-dysregulation, IR can also lead to the intracellular hydrolysis of Triglycerides (TGLs) and the hypersecretion of Low-Density Lipoprotein (LDL) Cholesterol (CHOL), further contributing to metabolic syndrome [102]. At the molecular level, the hypothesized embedded pathobiological mechanism, connecting all glucose metabolism dysregulation, energy bioavailability deficits, and lipid disturbances, involves (but is not limited to) the dysfunctional association of Phosphoinositide 3-Kinase (PI3K) with the Insulin Receptor Substrate (IRS), leading to the activation of downstream effectors, namely c-Jun N-terminal kinases (JNKs), inhibitory-kB kinase b (IKKb), mTOR/S6K, and extracellular signal-regulated kinases (ERKs) [103,104,105]. The loss of Insulin-R’s sensitivity has been associated with nutrient overload, systemic Fatty Acid excess, inflammation, hypoxia of adipose tissue, as well as endoplasmatic reticulum OXS [104,106]. As early as 1994, Pellerin and Magistretti coined the astrocyte–neuron Lactate shuttle model [107], introducing the notion of a close association between metabolism (glucose utilization) and neurotransmission (glutamatergic-Rs). Additional evidence further supporting the metabolism’s interplay with neurotransmission stems from the genetic field, whereby the gene *Disrupted-In-Schizophrenia 1 (DISC1)* has been speculated as a risk factor for both psychosis, based on its relevance to Dopamine dysregulation [108], and type 2 Diabetes (T2D), based on its relevance to glucose intolerance in transgenic mice [109].

The idea of a potential link between SCZ and hyperglycemia dates back to 1919 [110]. The application of an Insulin-induced hypoglycemic coma as a treatment in SCZ [111], along with the observation that higher doses of Insulin were needed to attain therapeutic hypoglycemia among psychotic patients, fueled the hypothesis of a higher prevalence of IR in SCZ [112]. At present, Diabetes prevails in SCZ, being twice as common as in general populations [113], thus leading to a shorter life expectancy of about 20 years [114].

Recent case studies supported the hypothesis of glucometabolic aberrations in psychosis as early in the course of the disorder’s evolution timeline as in the FEP stage, even prior to the commencement of medication [115,116,117,118,119,120,121] yet not always with consistency [122,123]. Later meta-analyses of glycemic indices in the FEP and FESCZ underlined IR and the oral glucose tolerance test aberration in early psychosis [124,125].

Additional findings—from different fields, utilizing varied biological samples, such as post-mortem brain tissue and DNA, and techniques such as PET, fMRI, metabolomic, and genetic ones—provide evidence for glucometabolic disturbances in chronically medicated and unmedicated psychosis cohorts [126,127,128,129,130,131]. The scheme of the ‘type 3 Diabetes’ notion could incorporate the above-mentioned findings, complemented by the ‘selfish brain’ theory [101], whereby the psychotic brain strives to compensate, due to energy deficiency, for the brain’s IR [132,133], finally affecting neurotransmission and consequently cognition [134].

The evidence for an inherently abnormal glucose metabolism in psychosis is prevalent in the prodromal stage as well. Indeed, relatives of FEP patients exhibited a higher prevalence of abnormal glucose tolerance against the HC [135,136]. However, glucose intolerance does not seem to show the capacity to predict transitions to SCZ [137]. On the contrary, findings from the UK ALSPAC Birth Cohort study support the hypothesis of IR being one of the basic players in the prospective interaction between inflammatory and glucometabolic abnormalities for the incidence of either psychotic experiences or disorders in adulthood [138].

Apart from the aforementioned, tentatively inherent, glucometabolic disturbances in psychosis, obesity and lipid dysregulation constitute parallel pillars of the overall metabolic burden in psychosis, leading to a shorter life expectancy of 20 years [114]. Indeed, patients with SCZ are at a fourfold increased risk of abdominal obesity and at a twofold increased risk of metabolic syndrome when compared with cohort-matched general population controls [139].

Typical lipid indices, such as total CHOL, HDL, LDL, and TGL, along with weight and BMI, appeared aberrant in early psychosis patients [140] and in chronic patients and were replicated by meta-analyses [139,141,142,143,144]. These studies attributed a central pathogenetic role to antipsychotic medication in the formation of metabolic burdens in psychosis. In an attempt to disentangle the medication factor from the metabolic signature in psychosis, case studies on drug-naive FEP patients suggested a lower plasma total CHOL and HDL compared to the HC [121,137,145,146] and elevated plasma LPC (16:0, 18:2, 18:1, and 18:0), serum long-chain Acylcarnitines (LCACs) (C14:1, C16, C16:1, and C18:1), serum CHOL Ester (CE) (15:0), CE (16:1), and Cholic Acid (CA) [147,148]. Notably, HDL could serve as a biomarker of SCZ [149], thus giving weight to the hypothesis that lipid abnormalities could not only emerge as medication effects but also may underlie psychosis’ pathobiology inherently [137,150]. Interestingly, despite the association of increased lipid clusters of polyunsaturated TGL (LC15,17,12) levels with medication, other TGL clusters (LCs 8, 9, and 13) were downregulated, thus suggesting a binary role (both aggravating and beneficial) of medication in the lipids’ signature in psychosis [151].

Complementary to the above evidence, studies involving UHR-P populations support the aberrant lipid metabolism hypothesis, based on lower HDL levels [146], as well as a lipid signature composed of ceramides (Cers) (d18:1/24:0), lysophosphatidlycholines (LPCs)(22:5), phosphatidlycholines (PCs)(38:4), PC(40:5), and PC (O-32:0), with the latter having the potential to not only differentiate UHR-P cohorts from the HC but also those UHR-P subjects who would convert to psychosis [152].

## 5. Neuroendocrine/Stress System

Another pillar of an organism’s response to psychological and physiological stressors is the hypothalamic–pituitary–adrenal (HPA) axis [153]. According to Hans Selye (1950) [154], one of the leading contemporary researchers of stress in the 1940s, the HPA axis’ response to stress, also known as the ‘General Adaptation Syndrome’, regulates body organs and systems’ function to ensure a psychological and physical balance (homeostasis). The Systemic/Adreno-Medullary (SAM) Sympathetic Nervous System complements the stress system, and when both axes are activated, Glucocorticoids (GCs) and Catecholamines (CAs) are secreted, impacting each other interchangeably, to maintain basal and stress-related homeostasis by suppressing inflammatory pathways [153]. Conversely, the absence of GC’s or GC-R’s reduced sensitivity—causing their incapacity to mount their biochemical and genetic effects [155], a state also known as ‘GC resistance’—deprives the organism of its anti-inflammatory shield, conferring vulnerability to unimpeded inflammatory processes, leading to neuronal death [156]. What is more, GCs exert multimodal effects in the brain’s microglia (pro- vs. anti-inflammatory), depending on the intensity/chronicity of the stressor and the timing relative to the stressful stimuli [11,157]. As far as the impact of GCs on neurotransmitters is concerned, evidence suggests that constitutively acute, moderate HPA axis activation causes the long-lasting potentiation of glutamatergic transmission in Prefrontal Cortex (PFC) pyramidal neurons, yet, the aberration (decrease/increase) in GC levels of a significant intensity and/or chronicity, due to impaired negative feedback, stimulates the striatal Dopamine release in both preclinical and clinical studies [158]. Other neuroactive steroids such as Dehydroepiandrosterone sulfate (DHEA-S) appear to be significantly related to dopaminergic neurotransmission as well [159].

This interaction between the neuroendocrine system, stress, and the central nervous system (CNS) contributes to an increased risk of psychosis, as illustrated by the neural diathesis–stress model of psychosis [158,160].The first studies on SCZ cohorts, through the neuroendocrine/stress angle, provided divergent results probably due to the sample heterogeneity in terms of the diagnoses, comorbidity, phase of illness, and exposure to medication [161,162]. Subsequent meta-analyses of cortisol in chronic SCZ cohorts suggested increased morning cortisol levels [163], as well as attenuated/blunted post-psychosocial/awakening cortisol responses [164,165,166].To further disentangle the HPA axis function in psychosis from chronicity and medication effects, research focused on the FEP stage, and the evidence favors the upregulation of blood cortisol at baseline as reviewed by Borges et al. (2013) [167] and Karanikas et al. (2014) [161], which was furthered by recent meta-analyses and replicated by studies on hair cortisol concentrations [168,169,170], yet there have not been consistent results [171]. As far as more dynamic cortisol characteristics are concerned, such as the Cortisol Awakening Response (CAR), FEP review studies suggested a blunted response [161,167]; recent meta-analyses [166,169] supported this outcome, thus strengthening the notion of a dysregulated HPA axis as early as the psychosis emergence. The potential implications of cortisol’s neurotoxicity, and hence its tentative involvement in psychosis’ pathobiology, can be indirectly inferred by longitudinal studies demonstrating antipsychotic medications’ beneficial effects on cortisol’s circulating levels (downregulation) [172,173], as well as on the Dexamethasone Suppression Test (DST) non-suppression rates (decrease) [174]. Complementarily, the HPA axis’s involvement in early psychosis pathobiology is evident via studies on prodromal subjects, whereby the baseline (unstimulated) salivary cortisol and the CAR appear to be upregulated [169,175,176,177] yet not consistently [169,178].

Intuitively, Labad 2019 conceptualized a varied neuroendocrine/stress system function along the psychosis evolutionary timeline, incorporating differential HPA axis perturbations at different stages [178].

As previously mentioned, in addition to cortisol, there are other neuroactive steroids with a regulatory role in stress responses; DHEA, with its potent anti-glucocorticoid activities, is one of these, appearing to play a neuroprotective role against corticosterone’s neurotoxicity [159,179,180]. Interestingly, a recent meta-analysis suggested increased DHEA-S (but not DHEA) levels in FEP cohorts [181]. In contrast, a recent quantitative review failed to replicate DHEA’s protective role in psychosis [182].

DHEA is a substrate in Testosterone (TEST) synthesis; hence, it is reasonable that research on cortisol’s role in psychosis pathobiology expanded on other neuroactive steroids, especially TEST [181]. As early as 2004, studies on TEST in drug-naive or drug-free male FESCZ cohorts started appearing and showed reduced total or free TEST levels [183,184,185], a lower free Androgen index [186], as well as TEST’s prophylactic role against the risk for psychosis. However, there was evidence of a different effect, with increased TEST in both FESCZ and UHR-P cohorts against the HC [187]. A recent meta-analysis incorporated the varied results relative to TEST along the course of psychosis, formulating a model of increased TEST as a response to stress early in the course of the disorder, which, along the timeline (episodic relapses), tended to abate, especially in males [181].

## 6. Multisystemic Interactions

As already stated in the introduction, any attempt to segregate the immune/redox/metabolic/neuroendocrine/stress fields is conventional and serves the purpose of better comprehension. The stress research scope has shown that these fields contribute to an organism’s homeostasis and health, yet inflammatory cascades are activated in cases of an increased allostatic burden [9,153,188,189]. The involvement of each one of the immune/redox/metabolic/neuroendocrine/stress fields in inflammatory processes, their consequent impact on the CNS, and the causality directions regarding their interplay—to finally form the pathobiological substrate on to which the symptom, of either a physical, emotional, or cognitive nature, will appear—represent labyrinths under constant research. Hitherto, the evidence from preclinical studies on the stress field suggests a tight interconnectedness of the elements representing these fields.

Preclinical research has shown that psychological and/or physiological stress can lead to chemokine and cytokine production, via DAMP effects on TLRs, with subsequent NLP-3 priming along with NF-kB and MAPK activation [190,191,192]. The aforementioned stage can sequentially be followed by the noxious inducible NOS (iNOS) activation and ROS/RNS upregulation [193], finally forming an activated vicious immune–redox loop. The substantiation of this loop is evidenced by increased pro-inflammatory cytokines, ROS, MAPKs, as well as the aberrant function of Akt and PI3Ks within the brain parenchyma and microglia in particular [65,194,195]. This way, despite constitutively serving homeostatic purposes, microglia can exert their inflammatogenic effects under intense/chronic stress conditions, culminating in neurotoxic ones [196]. This sequence may provide a translational perspective relative to psychosis’ hypothesized immune and redox pathobiology [65,197].

In addition to its impact on the redox system, the immune system exerts its effects on the neuroendocrine/stress system. Indeed, pro-inflammatory cytokines, under acute/moderate stress, activate HPA axis functions and lead to GC secretion, thus strengthening the organism’s anti-inflammatory potential, yet, under chronic/intense stress, they negatively impact GC-Rs, leading to the abatement of the HPA axis’s negative feedback capacity (resistance) [198]. Similarly to the HPA axis, the SAM axis can exert multimodal immunoregulatory effects [199], through Noradrenaline’s mediation of a1- and β-adrenergic-Rs activation via either the PI3K/MAPKs/NF-kB pathway [200] or the cAMP, ERKs-1/2, and p38 signaling pathways, in the absence of inflammatory stimuli [199], finally leading to immune activation. However, β2 adrenergic-Rs post-stress, via the PKA/PDE-4 pathway, exert an anti-inflammatory action [199], suggesting the multimodal interplay of the immune–neuroendocrine/stress systems. Following the same line of reasoning, as far as the immune–neuroendocrine/stress interplay in the context of the CNS is concerned, GCs can target microglia and exert either pro- or anti-inflammatory actions, depending on the timing and duration of the stressful insult [11], which adds to the versatile nature of the immune–neuroendocrine/stress interaction.

The immune system appears to be tightly interconnected with the redox and neuroendocrine/stress systems, as well as the metabolic one. Indeed, prenatal immune activation causes glucose metabolism dysregulation and abnormal ingestive behavior in peri-adolescence and leads to the adult onset of excess visceral and subcutaneous fat depositions [101,201,202]. Inversely, the deficiency of pro-inflammatory elements such as IL-1R [203], TNF-α [204], NF-κB [205], and JNK saved mice from IR and glucose intolerance [206]. Not only inflammatory cytokines but also Fatty Acid derivatives, such as ceramides and diacylglycerol, and ROS/RNS can negatively regulate Insulin signaling via the activation of Akt [104]. In a feed-forward fashion, Akt exerts a pro-inflammatory role by promoting the NF-kB translocation to the nucleus, thus strengthening the spiral inflammation–IR [207]. Taking into account Akt’s connection with Dopamine transporters pre-synaptically and its inhibition by D2-R activation [208], it is easily inferred that the immune–metabolic–CNS relationship is tight.

Despite the relative abundance of evidence favoring a close immune–redox interplay in the context of stress, when it comes to the aforementioned systems’ interplay in the clinical psychosis pathobiology context, only minimal evidence exists. Indeed, only a few studies referred to associations of the immune–redox components despite not constituting the primary research outcome. Specifically, Dwir et al. (2020), in their recent translational study, reported significant associations of immune alterations (NF-kB and MMP-9) with redox ones (RAGEs) [209]. These findings appear to align with findings from an earlier case study on drug-naive and drug-free SCZ cohorts [210], with the latter favoring significant correlations with inflammatory components (MMP-9) with both pro-oxidative parameters, MDA (positively), and anti-oxidant ones, TAS (negatively). Recent preliminary evidence from FEP cohorts also indicates a close immune–redox interplay, based on findings suggesting immune components’ (WBC, Neutrophils) [211] (IL-1, IL-4) significant positive association with redox components [212], either with pro-oxidative properties [total level of peroxides (TPX)] [147] or with anti-oxidant ones [SOD-1, CAT, ferric reducing anti-oxidant power (FRAP) [211], serum total anti-oxidant capacity (TAC) [147], taurine [212]]. A recent study underscored the importance of immune (TNF-a)–redox (MDA) interactions in increasing the risk for SCZ [213], hinting towards the hypothesized immune–redox interplay’s involvement in psychosis’ pathobiology.

Additional, yet indirect, evidence of a tentative immune–redox interplay in psychosis originates from FEP studies, with designs showing either a close correlation of each one of the immune (INF-γ, IL-8) and redox (NOS-1 genotype) parameters with a third parameter, deviated in psychosis (hippocampal volumetric integrity) [214], or synchronous alterations of each of the immune (IL-6, TNF-a, CRP) [215] (IL-1a, IL-6) [147] (NF-κB, Prostaglandin E2) [216] and redox (GSH, ferritin, TPX, Oxidative Stress Index, OSI) [147], (iNOS, NO2) [216] systems post-intervention (exercise) [215] (medication) [147,216], despite the lack of reports on direct correlations between them.

Only lately has the focus of research been directed towards the tentative immune–metabolic interplay in the context of psychosis. Prospective studies evaluated both immune and metabolic parameters as early in psychosis as in the FEP stage. The former (immune) component has involved the evaluation of CRP [116,138,217,218,219], cytokines, and chemokines [31,116,138,147,217,220]. In contrast, the latter (metabolic) component has involved the evaluation of the TGL, BMI, glycated hemoglobin (HbA1c), fasting glucose [217,218,219], HDL, LDL, total CHOL [217], Ghrelin, Glucagon, Insulin, Leptin [220], two-hour oral glucose tolerance test [116,123], ACs [31,147], and HOMA-IR [124]. The results are congruent with both immune and metabolic systems’ deviations from normality within the same cohorts, yet the reports concerning their interplay are only sporadic and indirect. Specifically, the findings suggesting the most substantial evidence for an immune–metabolic interplay in psychosis come from Russel et al. (2015) [219], presenting not only a significant, positive correlation between the increases in hsCRP and TGL in medicated FEP cohorts but also the capacity of the hsCRP increase to serve as a predictor of TGL’s upregulation in the short term, independently of increases in weight or of baseline inflammatory states. Similarly, Perry et al. (2019) [138], utilizing data from the ALSPAC study, proposed that immune dysfunction could also be upstream of dysglycaemia in its relationship with psychosis, based on the significant interaction between IL-6 (age 9 y) and IR (age 18 y) for psychotic experiences and Psychotic Disorder (age 18 y). Additional, yet indirect, evidence of the immune–metabolic interplay in the context of psychosis was provided by a recent study showing the capacity of hsCRP [218], clustered with TGL and the BMI, to predict the negative outcome of the treatment response. In contrast, another study [217], whose design was based on the evaluation of the Allostatic Load (AL) index, encompassing (but not limited to) immune and metabolic parameters in an FEP cohort, failed to attribute a predictive capacity of either the treatment response or functional remission of psychosis to the AL.

Studies from the genetic field offered additional indirect evidence of potential interactions between the immune–metabolic systems in psychosis.

Liu et al. (2013) [221] demonstrated that immune and metabolic pathways, specifically related to the *TNF* and *Akt* genes, can represent shared risk factors for both psychosis and Diabetes. Additionally, a subsequent study found a significant correlation between the IL-6 and *Akt* gene during the FEP [222]. Given Akt’s crucial role in cerebral Insulin signaling [127,128], the findings from these genetic studies suggest a significant interaction between immune and metabolic systems in the context of psychosis. All in all, it appears that the immune–metabolic interplay research in the psychotic context is starting to grow, with only minimal evidence hitherto hinting at an intersystemic overlap concerning the psychosis pathobiology.

The evidence regarding the simultaneous evaluation of redox and metabolic parameters within the same psychotic cohort seems scattered and inconsistent. The earliest evidence suggestive of a common dysfunction of redox and metabolic systems in psychosis stems from post-mortem studies on SCZ patients, indicating a decrease in the volume density and the count of mitochondria in oligodendroglial cells in the caudate nucleus and prefrontal area [223,224]. Mitochondria are known pillars of ATP production via OXPHO, as well as ROS/RNS production as by-products of electron transportation chain functions [225]; hence, they are at the crossroads of both redox and metabolic systems. A later study on SCZ cohorts, using a novel 31P-MT-MRS approach, underlined the significant negative association of a reduced pH (indicating buildup of Lactate, possibly due to slowed ATP synthesis rates, as result of downregulated OXPHO) with an increased BMI (index of abnormal peripheral metabolism) [226]. The hypothesis for a close redox–metabolic system interplay is furthered by studies on drug-naive FESCZ, suggesting a significant negative association between the BMI and TAS, both at the baseline and post-medication [227]. The TASxBMI interaction appeared to confer predictive potential on the outcome of negative symptom improvement [228]. In a similar cohort, IR (increased) on one hand and pro- and anti-oxidant aberrations (increased serum GSSG/NO pro-oxidants levels and decreased serum anti-oxidant SOD levels, respectively) on the other were associated with the severity of the cognitive impairment, constituting another indirect piece of evidence for the redox–metabolism interplay in psychosis, yet there is no report on any direct associations among redox and metabolic parameters [229]. Notably, the genetic field further supports the notion of the redox–metabolic system interplay based on findings, suggesting that 90% of the genes related to energy metabolism and OXPHO are altered in SCZ patients [230]. Chromosome 1q21–25 can constitute a promising region at the crossroads of SCZ, T2D, and redox (the *CAPON* gene encodes an adapter protein targeting neuronal NOS) [231,232].

Only recently (in the last decade) has the research community started studying parameters from the above-mentioned fields simultaneously. This attempt incorporated different techniques and methods, namely proteomics [90], metabolomics [148], genetics [233], and AL evaluations [217]. One research group expanded their research depth and width in the immune scope by using multiple analytes to gauge as many immune parameters as possible in blood and CSF with DNA-labeled antibodies and real-time PCR techniques [234]. In contrast, other research groups expanded their research beyond the fields covered in this review (immune/redox/metabolic/neuroendocrine/stress) to other scopes, such as the brain structure, neurophysiology, and neurochemistry [235]. Despite this synchronous evaluation of parameters representing different pathobiological fields in the context of psychosis, again, the reports relative to their in-between associations are minimal. Specifically, as early as 2014, van Beveren et al. [236], recruiting subjects from the Dutch’ Genetic Risk and Outcome in Psychosis’ (GROUP) project, showed deviated parameters in the metabolism (Insulin, C peptide increase), immune (T-lymphocyte-secreted protein I 309), as well as neuroendocrine/stress fields (Growth Hormone decrease) in SCZ subjects. Later on, Berger et al. (2018) [217] gauged the AL index in both chronicity and the FEP, encompassing cardiovascular (i.e., heart rate, blood pressure), neurohormonal (cortisol), metabolic (TGL, CHOL, BMI), and immune (CRP, cytokines) parameters; the results converged towards increased ALs. However, none of the studies included reports on tentative associations among the different pathobiological fields. Subsequently, a proteomic study of 4136 nonredundant proteins in a drug-naive FEP cohort [90] proposed altered (heightened) rates of function within each of the immune (C1R and five regulatory proteins cascade immune system, including FH, FI, CR1, C4BPA, and CD59), redox [TRX Reductase1, TRX (TXNDC5 and TXNDC17), methionine sulfoxide reductase (MSRA), and Peroxiredoxin (PRDX4 and PRDX5) anti-oxidants] and metabolic systems [proteins participating in glycolysis, Tricarboxylic Acid (TCA) cycle, OXPHOS and ATP synthase complex function]. Again, no report was produced on associations among parameters representing the different systems. Another field where the multisystemic nature of the psychosis etiopathology was studied is the genetic one. Indeed, a recent umbrella review of meta-analyses and Mendelian randomization studies on SCZ provided evidence for immune (TNF-a) as well as redox parameters (plasma TAS) that may play a role as risk factors [233]. Further, yet indirect, evidence favoring a potential multisystemic interaction in the context of psychosis originates from studies developing algorithms for either a psychosis diagnosis against normality [148,237], the differentiation of the psychotic substrate in affective disorders [238], or the prediction of the conversion to psychosis from prodrome [35]. These studies encompassed parameters from the neuroendocrine/stress (cortisol increase) [148], metabolic [LDL, Cer (d18:1/22:0) (downregulation), AC C2 (downregulation) [148], sum of Creatine and Phosphocreatine, glucose, Insulin [237,238]], immune [(IL-1b, IL-7, IL-8, IgE [35], Reactive-Lymphocytes [237], WBC [238])], and redox fields [(Taurine [237], Uric Acid [238], MDA [35])]. However, there were no reports on the tentative associations among parameters from the involved fields.

Moreover, limited and inconsistent evidence emerged when a study focused on the tentative association of pillars (brain structure, neurophysiology, neurochemistry) of the CNS function with pillars of non-CNS (immune, cardiometabolic, and HPA axis) functional systems in the context of psychosis, despite the evidence for the aberrant function of each system separately [235]. Again, the redox system was not included in this meta-review.

## 7. Discussion

During the last few decades, evidence from biological psychiatry has accumulated exponentially, suggesting the involvement of several systems, namely the immune, redox, neuroendocrine/stress, and metabolic ones, in the alteration of brain structures as well as functions in the context of psychosis. The evidence appears strongly suggestive of an inherent participation of each one of the aforementioned systems in the psychosis pathobiology, based on findings indicative of their aberrant function as early as the FEP stage and even before that (prodrome). However, the exact etiological sequence and interplay among the deviated systems’ functions and the consequent effect on the brain constitute a field of painstaking research, the elucidation of which appears to have only started emerging.

A recapitulation of the accumulated knowledge regarding each reviewed field will follow, beginning with the immune component. Indeed, there is relatively strong evidence favoring the activation of pro-inflammatory immune processes, implicating both innate and Thelper-1 immunity, as assessed mainly through cytokine levels [28,29]. Regarding the activation of the anti-inflammatory immune system, the evidence appears more controversial, with findings supporting either the non-activation [28,29] or deviation of varied anti-inflammatory cytokines (IL-1RA, IL-10) [17,239], (TGF-β) [30], (IL-4) [31]. The scene appears far more complex given that cytokines such as IL-6 can perform both pro-inflammatory and regulatory roles [32]. In addition, there is speculation that the immune–inflammatory response system (IRS)/compensatory immune–regulatory reflex system (CIRS) ratio plays a regulatory role in immunity rather than the absolute counts of each cytokine [32]. Similarly, the redox system appears to inherently be involved in the psychosis pathobiology, as evidenced by the elevation of pro-oxidant toxic by-products, not only in chronicity [70,71] but also in the initial full-blown stage [15,74,75] and even before that [35]. Moreover, the anti-oxidant defenses are also perturbed in chronicity [15,76], the FEP [15,16,73,78,240], as well as the prodromal stage [91,92], collectively furthering the evidence of increased OXS in psychosis. Decreased GSH levels appear to constitute the most reproducible finding in relation to anti-oxidant elements throughout the evolutionary path of psychosis, replicated in different biological samples in both chronicity [81,82,83,84] and the FEP stage [83,85,86,87,88]. When it comes to the function of the primary anti-oxidant enzymatic system, namely the TRX system, the evidence favors a hypothesis of an increased TRX system function as soon as the initial demonstration of full-blown psychosis, possibly due to high energy demands [8]. In contrast, preliminary evidence suggests a decreased TRX system function when chronicity evolves [241], compatible with downregulated energy production and consumption.

The aforementioned tentative bioenergetic deficit in psychosis, intuitively, leads the reader to expect glucoregulatory aberrations since glucose is the primary energy cell supplier. Indeed, there is growing evidence for glucose metabolism aberrations in psychosis as early in the course of the disorder’s evolution timeline as the FEP stage, even before the commencement of medication [115,116,117,118,119,120,121]. This evidence of the peripheral dysregulation of glycemic indices (high fasting blood glucose, impaired HOMA-IR, and glucose tolerance test) coupled with the evidence of central (cerebellar) Insulin signaling and consequent glucose metabolism impairments recapitulates the notion of generalized glucose metabolism impairment, thus leading to the bioenergetic deficit in psychosis [126,128,129,130,131].

Lipids are another fundamental pillar of the energy repository and a structural constituent of the cell membrane. Thus, intuitively, it would be expected that lipids—apart from their profound role in increasing the metabolic burden, leading to cardiovascular disease, conditions known for their high prevalence in psychosis [139]—may relate etiologically to the latter’s pathobiology. Indeed, lipid metabolism indices were aberrant in psychosis, even before medication commencement [121,137,145,146]. Interestingly, the most consistent lipid dysregulation findings at the initial stage of psychosis were a lower total CHOL and HDL [121,145]. Recent lipidomic techniques complement the notion of lipid abnormalities in psychosis, with a signature entailing LPCs, Cers [152], as well as clusters of TGL with differential unsaturated carbon–carbon bond numbers [151,152]. Notably, the findings provide insight into the lipid perturbations evaluated in the periphery (blood biological samples), which do not necessarily correspond to central (brain) aberrations.

Neuroendocrine/stress hormones complement the puzzle of potential players, forming the pathobiological substrate on which psychosis evolves. Indeed, the evidence supports the idea that the HPA axis function starts being deviated as early as the initial FEP and even before that. Evidence of the upregulation of the baseline cortisol secretion and blunted cortisol reactivity post-stressor [161,242] has enriched the neural diathesis–stress model of psychosis [158], thus suggesting altered circulating cortisol’s setting point as well as GC-R (dys)function (resistance). The evidence concerning HPA axis dysfunction at prodrome is more ambiguous [178,242], but this ambiguity lessens with the proximity to the initial full-blown psychosis.

When it comes to the interplay of these players (immune/redox/metabolism/neuroendocrine/stress) in the context of psychosis, the scene becomes more complex. Despite the preclinical evidence from the stress field of the close interconnectedness among those scopes, the respective findings in clinical settings involving patients with psychosis range from very little to none. Only recently have there been sporadic studies incorporating the simultaneous evaluation of several parameters of the immune/redox/metabolism/neuroendocrine/stress fields in their design, yet reports on their associations are minimal.

The interactions between immune and redox systems in psychosis show promise, even though these findings were not the primary outcomes of the studies. Indeed, various immune parameters (cytokines, WBC, NF-kB, MMP-9, CRP, Prostaglandines) were evaluated along with redox parameters of either the pro-oxidant (MDA, TPX) or the anti-oxidant (TAS, SOD-1, CAT, FRAP, TAC, Taurine) clusters in cohorts with the FEP and SCZ, with a differential medication status (medicated, drug-free, or drug-naive) [147,209,210,211].

As far as the immune–metabolic interplay in the context of psychosis is concerned, it appears hitherto unexplored, with only minimal evidence. The studies incorporating an evaluation of both fields in their design gauged various immune parameters (CRP, cytokines, chemokines) as well as metabolic ones (TGL, BMI, HbA1c, fasting glucose, HDL, LDL, total CHOL, Ghrelin, Glucagon, Insulin, Leptin, two-hour oral glucose tolerance test, Acs, and HOMA-IR), based on blood samples, in FEP cohorts [116,138,217,218,219]. The preliminary evidence suggests a close interplay between the change in immune (CRP)–metabolic (TGL) parameters at the FEP stage under medication conditions [219]. What is more, in terms of the etiological sequence, it appears that immune dysfunction may lie upstream of metabolic perturbations in their relationship with psychosis [138].

Despite the difficulty of discerning the redox/metabolic fields, due to their functional interdependence, preliminary evidence suggests their close interaction in the context of psychosis. The BMI, as an index of abnormal peripheral metabolism, is omnipresent in studies, suggesting associations with redox constituents, such as Lactate (increase), indicating reduced OXPHO [226] and TAS [227,228]. In addition, Insulin might be closely related to serum levels of GSSG, NO pro-oxidants, as well as SOD anti-oxidants [228]. Again, there was no report on the direct associations of Insulin with the aforementioned redox parameters.

To make matters more complex, a recent scoping review from our team that aimed to search for any evidence relative to cortisol–cytokine interactions in FEP and UHR-P cohorts provided counterintuitive results [243]. Indeed, the findings for the primary outcome were either negative or not reported, despite each of the studied fields [immune(cytokines)/neuroendocrine(cortisol)] separately exhibiting evidence of dysregulation in early psychosis.

The reader needs to bear in mind that the segregation of the reviewed scopes, namely immune/redox/metabolic/neuroendocrine/stress fields, is somewhat schematic and meant to serve the practicality of the review and the comprehension of the mechanisms. Figure 1 schematically depicts the author’s hypothesis relative to the multisystemic involvement in the pathobiology of psychosis, based on both preclinical findings in the context of stress and clinical ones with psychosis cohorts.

In reality, accumulating knowledge requires a close interdependency among the scopes, tightened as a Gordian knot [20]. When functioning optimally, the systems preserve cellular homeostasis and hence the organism’s health. However, in cases of failure to preserve the intersystemic balance, the AL accumulates, resulting in cellular inflammation and leading the organism to develop disease [9,151,187].

Certain domains, particularly the kynurenine pathway, are involved in regulating intersystemic interactions. Inflammatory cytokines enhance the conversion of tryptophan to kynurenines. Kynurenic acid (KYNA) affects neurotransmitters such as Glutamate, Dopamine, Acetylcholine (Ach), and GABA by antagonizing α7 nicotinic Ach–Rs. Additionally, KYNA blocks NMDA-Rs on PVIs, disrupting midbrain dopaminergic signaling [244]. This creates a vicious cycle of inflammation, metabolism dysregulation (tryptophan to kynurenines), and neurotransmitter imbalance. This underlying vicious pathobiological mechanism may explain the close association of psychosis with depression.

All in all, it appears that the literature lacks evidence (or it is only minimal) concerning the intersystemic immune–redox–metabolic–neuroendocrine/stress interactions in psychosis. This lack may be explained by researchers not having found significant ‘positive’ results, hence omitting to report them, or not having gauged the interactions. The heterogeneity of cohorts’ diagnoses combined with the relatively small groups of participants constitute factors that could explain the difficulty of detecting aberrant pathobiological multisystemic patterns. Indeed, the FEP diagnosis encompasses potentially different diagnostic entities, which will variably evolve along the disorder’s timeline. The same obscurity characterizes the UHR-P state, composed of different prodromal phenotypes with a varying progression to full psychosis. The medication effects on the reviewed systems’ function—along with the heterogeneity of the study design, gauging varied components of the respective systems—can explain the lack of findings relative to the systemic dysfunction or intersystemic associations.

Moreover, the varied evaluation of either levels or functions of immune/redox/metabolic/neuroendocrine/stress parameters can further fuel the heterogeneity in design levels, hence the difficulty in completing the pathobiological puzzle. Specifically, one anti-oxidant’s level does not necessarily reflect its functional capacity. The same obscurity characterizes the study of the stress hormones’ function, where instant snapshots of their levels do not necessarily reflect the diurnal total secretion nor their diurnal fluctuation, not the least the affinity with receptors and the consequent regulatory feedback capacity. The field may become far more complex, considering that different stressors (pharmaceutical–psychosocial–environmental) may elicit different immune/redox/metabolic/neuroendocrine/stress responses. The variation in the biological tissues (plasma, serum, saliva, hair, CSF) where the systems’ components are evaluated may further explain the variance in the results. Last but not least, the misuse of psychotropic substances like marijuana further hampers any attempt to disentangle the multisystemic directionality, as recent studies are suggestive of the latter’s pro-inflammatory/oxidative/gluco-dysregulatory/hyper-cortisolemic effects [245,246,247]. Similarly, other recreational substances (curcumin, green tea, ginseng) have been suggested to interfere with the immune/redox/metabolic/neuroendocrine/stress systems, in an anti-inflammatory fashion; hence, they may possibly be prophylactic against psychosis, whereas others (ayahuasca, kratom) act in an opposite way [248,249,250].

## 8. Future Directions

It becomes apparent that new research avenues are evolving in the attempt to unravel the complex pathobiological substrate in psychosis, where the evidence favors its multisystemic nature. It appears imperative that researchers should not only aim to gauge each system’s parameter levels, function, and activation state in various tissues both peripherally (circulation) and centrally (brain) but also the intersystemic associations in large, well-controlled, drug-naive cohorts, prospectively, with the initial recruitment as proximal to an initial diagnosis and even before that (prodrome) if possible. Researchers should not only take into consideration the participants’ psychopathology (positive, negative, cognitive) and the disorder’s phase (active, relapse, refractory) but also the underlying general medical condition (forming the basis of exclusion criteria). Authors and readers should also keep in mind, before extrapolating any presumptions, that some biological components, such as IL-6, exert a pleiotropic role, participating in pro-inflammatory and anti-inflammatory processes. Translational studies need to exercise caution before projecting findings from animal studies to humans. A deeper understanding of the role that various biological parameters play in the underlying pathology of psychosis could advance the field of precision psychiatry. This could lead to the development of new pharmacological agents that specifically target the immune, redox, metabolic, and neuroendocrine/stress systems. As a result, it would enable more effective treatments for cases of psychosis that are resistant to traditional therapies [251]. In addition, everyday lifestyle practices—such as a diet rich in omega-3 Fatty Acids and herbs with anti-oxidant properties, gut health-promoting activities, regular exercise, and mindfulness techniques—are beginning to show their potential benefits for maintaining the balance in these systems [215,248,249,252]. Together, these approaches could enhance the toolkit available for treating psychosis.

## 9. Conclusions

This narrative review of an exploratory nature aimed to collect evidence of the interactions between the immune–redox–metabolic–neuroendocrine/stress systems in order to form a pathobiological framework for psychosis. Despite abundant preclinical evidence of the intersystemic interplay in stress, findings in clinical settings with psychosis patients remain minimal. The evidence relevant to each of the reviewed systems shows functional aberrations in psychosis, yet when it comes to the evidence concerning the intersystemic interplay, it appears scarce, contradictory, and even counterintuitive. Hence, the challenge of unraveling the complexity of the intersystemic interplay in the context of psychosis is imperative. The author hopes that this review will establish a framework for the further exploration of the association characterizing how the pathobiological ‘players’ orchestrate the etiopathology involved with psychosis.

## Figures and Tables

**Figure 1 neurosci-06-00099-f001:**
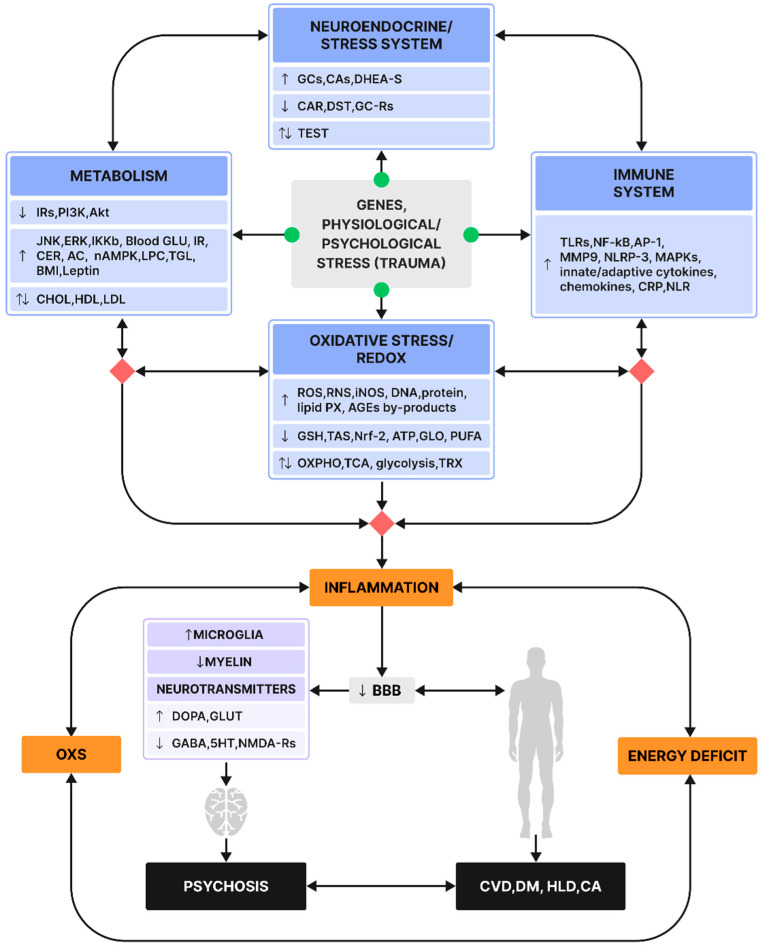
Schematic illustration of the suggested biological mechanisms underlying the pathobiology of psychosis as well as comorbid systemic disorders. Particular emphasis is placed on the interactions between the immune, redox, metabolic, and neuroendocrine/stress systems, based on both preclinical and clinical findings. Green color: starting point; red color: systemic interactions. Arrows within boxes: ↑ increase in concentration/function, ↓ decrease in concentration/function, ↑↓ evidence of both increase/decrease in concentration/function. Arrows linking the boxes: onward arrow →: feedforward causal effect, left right arrow ↔: bidirectional relationship. Abbreviations: Acylcarnitines (ACs), Activator Protein (AP), 5 Adenosine Monophosphate-Activated Protein Kinase (AMPK), Advanced Glycation End-Products (AGEs), Astrocytic AMPK (aAMPK), Blood–Brain Barrier (BBB), Cancer (CA), Cardiovascular Disease (CVD), Catecholamines (CAs), Ceramide (CER), Cholesterol(CHOL), c-Jun N-Terminal Kinase (JNK), Cortisol Awakening Response (CAR), C-Reactive Protein (CRP), Dehydroepiandrosterone Sulfate (DHEA-S), Dexamethasone Suppression Test (DST), Diabetes Melitus (DM), Dopamine (DOPA), Extracellular Signal-Regulated Kinase (ERK), Gamma-Aminobutyric Acid (GABA), Glucocorticoid-Receptors (GC-Rs), Glucose (GLU), Glutamate (GLUT), Glyoxalase (GLO), Hydroxytryptamine (HT), Hyperlipidemia (HLD), Hypothalamic/Pituitary/Adrenal (HPA), Inducible iNO Synthase (iNOS), Inhibitory-kB Kinase b (IKKb), Insulin Receptors (IRs), Lysophosphatidyloholine (LPC), Metalloproteinase (MMP), Mitogen-Activated Protein Kinases (MAPKs), Neuronal AMPK (nAMPK), Neutrophile Lymphocyte Ratio (NLR), Nuclear Factor-k Beta (NF-κB), Nucleotide-Binding Domain, Leucine-Rich-Containing Family, Pyrin Domain-Containing-3 (NLRP-3), N-Methyl-D-Aspartate Receptors (NMDA-Rs), Nuclear Factor E2-Related Factor (Nrf), Oxidative Phosphorylation (OXPHO), Peroxide (PX), Phosphatidylinositide 3-Kinase (PI3K), Polyunsaturated Fatty Acid (PUFA), Reactive Oxygen/Nitrogen Species (ROS/RNS), Serine/Threonine-Specific Protein Kinase (Akt), Total Anti-Oxidant Status (TAS), Testosterone (TEST), Thioredoxine (TRX), Toll-Like Receptors (TLRs), Tricarboxylic Acid (TCA), and Triglycerides (TGLs).

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
