# Peer review of "The Pathobiological Underpinnings of Psychosis: From the Stress-Related Hypothesis to a Multisystemic Approach"

_neurosci, 2025, doi:10.3390/neurosci6040099_

Round 1

Reviewer 1 Report (Previous Reviewer 2)

Comments and Suggestions for Authors Accept.  

Author Response

Comment: Acceptance

Reply: The Author wishes to thank Reviewer 1 for their time, effort, and acknowledgment of soundness as well as the suggestion of acceptance.  

Reviewer 2 Report (Previous Reviewer 3)

Comments and Suggestions for Authors

The authors presented an article titled “Pathobiological underpinnings of Psychosis; from the Stress-Related Hypothesis to a Multisystemic Approach” aims to explore the assumption that these biological fields can modify their structural and functional states under chronic or intense stress, leading to alterations in pathobiological pathways.

The author is encouraged to address the following concerns:

1. The author attempts to provide a concise introduction as a platform for the manuscript and would like to have 1 sentence with the prevalence of psychosis in the world or at least in Portugal in the first 2 sentences.

2. Please also provide manuscript is so important as there are people who despite traditional methods of psychopharmacology remain treatment resistant.

3. The immunity and redox sections are well developed and sound in text. 

4. Figure 1 is well developed. Please provide a higher resolution image.

5. Based on your findings, what avenues of clinical therapeutics can we re-purpose to augment the current medications we use in psychiatry that may potentially modulate the inflammatory pathway?

Author Response

Comment  1. The author attempts to provide a concise introduction as a platform for the manuscript and would like to have 1 sentence with the prevalence of psychosis in the world or at least in Portugal in the first 2 sentences.

Response 1. The Author wishes to thank Reviewer 2 for their time and effort in providing feedback on the current manuscript under review. Specifically, the Author wishes to express his gratitude for the Reviewer’s high score of 4/5 stars in their evaluation of the manuscript. Please note that the yellow highlight represents changes post 1st revision, the green highlight post 2nd revision, and the blue highlight post 3rd revision.  

It has been added “Psychosis constitutes one of the most debilitating conditions of human health pathology, with an estimated 4.6 and 7.49 per 1000 persons pooled median global prevalence, and median lifetime prevalence, respectively [1]”.

Comment 2. Please also provide manuscript is so important as there are people who despite traditional methods of psychopharmacology remain treatment resistant

Response 2. The Author wishes to thank Reviewer 2 for their thoughtful remark.  Now it has been added,

 “A deeper understanding of the role that various biological parameters play in the underlying pathology of psychosis could advance the field of precision psychiatry. This could lead….. As a result, it would enable more effective treatments for cases of psychosis that are resistant to traditional therapies [251]. ….. Together, these approaches could enhance the toolkit available for treating psychosis.”        

Comments 3. The immunity and redox sections are well developed and sound in text. 

 Response 3. The Author expresses his gratitude to the Reviewer for their acknowledgment of the manuscript’s scientific and writing soundness.

Comment 4. Figure 1 is well developed. Please provide a higher resolution image.

 Response 4. The Author has now provided a higher resolution image.

Comment 5. Based on your findings, what avenues of clinical therapeutics can we re-purpose to augment the current medications we use in psychiatry that may potentially modulate the inflammatory pathway?

Response 5. The Author is feeling thankful to Reviewer 2 for their knowledgeable remark. The text has been dispersed with hints in that direction. In particular, it has been stated

 “even antidepressant medications, through their action on the STAT/JAK pathway, which is thought to participate in the pathobiology of psychosis, may have an adjunctive therapeutic role against psychosis [47].”, 

“Beyond the conventional limits of psychotropic medications, other agents with immunomodulating properties, such as NSAIDs, COX inhibitors, minocycline, and N-Acetylcysteine (NAC), have shown beneficial effects in early psychosis as adjunct agents [48].”

“Golden standard methods for refractory states in psychosis, such as Electroconvulsive Therapy (ECT), have only recently started being associated with immunomodulating effects, as evidenced by IL-6 and CRP decrease [49].”

“DHEA, with its potent anti-glucocorticoid activities, is one of these, appearing to play a neuroprotective role against corticosterone's neurotoxicity [157,177,178].”

TEST’s prophylactic role against the risk for psychosis.”

other recreational substances (Curcumin, green tea, ginseng) have been suggested interfering with the immune-redox-metabolic-neuroendocrine/stress systems, in an  anti-inflammatory fashion, hence possibly prophylactic against psychosis”

A deeper understanding of the role that various biological parameters play in the underlying pathology of psychosis could advance the field of precision psychiatry. This could lead to the development of new pharmacological agents that specifically target the immune, redox, metabolic, and neuroendocrine/stress systems.”

In addition, everyday lifestyle practices—such as a diet rich in omega-3 fatty acids, herbs with antioxidant properties, gut health, regular exercise, and mindfulness techniques—are beginning to show their potential benefits for maintaining balance in these systems [248,249,251,252]. Together, these approaches could enhance the toolkit available for treating psychosis.”     

Last but not least, the Reviewer is kindly requested to take into consideration that throughout the revision process (two major revisions and one minor), the Author was prompted by the previous Reviewers to decrease the content of the initial sections of the manuscript; hence, the final version has undergone major alterations towards the direction of length decrease. 

The Author once more expresses his gratitude to Reviewer 2 for their time and effort.  

This manuscript is a resubmission of an earlier submission. The following is a list of the peer review reports and author responses from that submission.

Round 1

Reviewer 1 Report

Comments and Suggestions for Authors

This is an interesting review that tried to summarize the complex multi-systematic involvement of inflammation, oxidative stress, metabolism and neuroendocrine systems and their "interactions" in the pathogenic processes and pathways of psychosis from stress, and perhaps genetic risks. The review has cited over 200 references with evidence from each aspect. However, the review remains on a descriptive/narrative mode without a clear hypothesis behind such complex manifestation at a multi-systematic level. Would all the involved systems be parallel and their association/implication in psychosis at the same level? or they have underlying causal or consequential relationships? Plus, the 2nd part of Immune-redox-metabolic-neuroendocrine/stress and Central Nervous System (CNS) is too broad; and not directly focused on pathogenesis and pathways of psychoses.  Using illustration to summarize the key concepts of the review is good but the only figure is in poor quality, both in contents (eg., unclear/confusing concepts) and in illustrative design.  

Author Response

Reviewer 1

This is an interesting review that tried to summarize the complex multi-systematic involvement of inflammation, oxidative stress, metabolism and neuroendocrine systems and their "interactions" in the pathogenic processes and pathways of psychosis from stress, and perhaps genetic risks. 

Reply: The Author would like to first acknowledge and thank Reviewer 1 for the time and effort dedicated to the review process. In addition, the Author expresses his thankfulness to Reviewer 1 for acknowledging that this review is interesting.

The review has cited over 200 references with evidence from each aspect. 

Reply: The Author wishes to thank Reviewer 1 for acknowledging that this Review incorporated more than 200 references (275 references specifically in the 1st draft), indicating the wide extent of the covered scope, hence the high degree of difficulty in performing each of the manuscript’s stages, the data collection, critical appraisal, synthesis, and writing   

However, the review remains on a descriptive/narrative mode 

Reply:   

The Author thanks Reviewer 1 for their feedback. Indeed, the manuscript emphasizes that the narrative mode has an exploratory character, as noted in phrases like “narrative review with exploratory character” and “This narrative review of exploratory nature aimed to collect evidence of the interactions between the immune, redox, metabolic, and neuroendocrine/stress systems to form a pathobiological framework in psychosis.”

However, the Author intends to convey that this review goes beyond simply describing data. The motivation to undertake this review stemmed from the growing evidence across various fields of basic medicine showing the involvement of different biological factors in the development of psychosis. Despite this, there have been few reports addressing the interactions between these systems. Basic research on stress fields offers translational potential for understanding how complex systems operate, yet the existing literature is largely obscure and lacks data on these intersystem interactions.

With this context in mind, the Author has organized extensive literature, declassifying and reclassifying the systems with a specific focus on their interactions. Throughout the manuscript, the author demonstrates an effort to categorize evidence regarding how a system functions, whether toward a particular pathway or in the opposite direction, influenced by various factors. There are several instances where the author highlights that the directionality of these systems can lead to the formation of vicious cycles or feed-forward cascades with varying outcomes.

These conclusions arise from the Author's critical evaluation of the highly heterogeneous and extensive literature. The culmination of this critical approach is an attempt to create a figure illustrating a hypothesis about how all the variables may (dys)function collectively. This suggests an approach that is not limited to a mere description of existing data, despite the Author's characterization of the review as “narrative with exploratory character.”

without a clear hypothesis behind such complex manifestation at a multi-systematic level. 

Reply: The Author wishes to thank Reviewer 1 for his/her thoughtful remark. 

Now the below statements were added

The hypothesis underlying this review is that the various scopes discussed are closely interconnected and exhibit different directions of causality. It is proposed that the components within each field not only relate to other components in the same field at different levels—through feedforward, mediating/moderating, or bottom-up/top-down processes—but also interact with components from other fields under review in a similarly multi-directional manner. This review aims to explore the assumption that these biological fields can modify their structural and functional states under chronic or intense stress, leading to alterations in pathobiological pathways. These changes may impact and be influenced by brain structures, thereby linking to the pathobiology of psychosis.”

The Figure1 depicts schematically the author’s hypothesis relative to the multisystemic involvement in the pathobiology in psychosis, based on both  preclinical findings in the context of stress and clinical ones with psychosis cohorts” 

Would all the involved systems be parallel and their association/implication in psychosis at the same level? or they have underlying causal or consequential relationships?

Reply: The Author acknowledges the Reviewer’s 1 insightful question -statement.

 It has already been answered that  the Author has organized extensive literature, declassifying and reclassifying the systems with a specific focus on their interactions. Throughout the manuscript, the author demonstrates an effort to categorize evidence regarding how a system functions, whether toward a particular pathway or in the opposite direction, influenced by various factors. There are several instances where the author highlights that the directionality of these systems can lead to the formation of vicious cycles or feed-forward cascades with varying outcomes.

The culmination of this critical approach is an attempt to create a figure illustrating a hypothesis about how all the variables may (dys)function collectively, and their directionality.

 Plus, the 2nd part of Immune-redox-metabolic-neuroendocrine/stress and Central Nervous System (CNS) is too broad; and not directly focused on pathogenesis and pathways of psychoses.

Reply:  

The author wishes to extend gratitude to Reviewer 1 for their valuable input and takes their comments seriously. As outlined in the title of this manuscript under review, "The Labyrinth of the Immune-Redox-Metabolic-Neuroendocrine/Stress Multisystemic Interplay: From Stress to Psychosis Pathobiology," this review aims to evaluate the existing evidence regarding the involvement of these various factors in both stress and psychosis. Based on over 20 years of expertise in the field, the author’s prior impression is that systematic research on the role of each one of the reviewed systems in psychosis has only emerged over the last decade or two. Nonetheless, references detailing how these systems are interconnected in forming the substrate of psychosis remain minimal, if they exist at all. This observation motivated the author to embark on this research journey. Historically, the theoretical framework explaining how these systems function has primarily been based on preclinical studies related to stress. For a long time, stress has been viewed as a critical catalyst impacting brain structures and other bodily organs. The neural diathesis-stress model, conceptualized by Walker and Diforio in 1997, encapsulates the close relationship between stress, brain dysfunction, and the likelihood of experiencing psychosis. Since then, researchers have systematically explored the neurobiological foundations of psychosis, leading to an era in which major fields of medicine—such as metabolism, immunity, redox biology, and neuroendocrinology—are under constant investigation to better understand the pathobiological substrate of psychosis. As knowledge of each proposed pathobiological pillar expands, novel factors, including neurosteroids, gonadal steroids, and novel metabolic factors, are continually added to the discussion surrounding the underpinnings of psychosis. Presently, researchers are on the cusp of an era in which a multisystematic approach to understanding pathobiology is gaining prevalence. Consequently, readers of academic papers are presented with a wealth of information covering various scopes. This situation has encouraged the author to integrate knowledge from multiple areas believed to contribute to the pathobiology of psychosis. Starting with stress as a foundation, the author aims to guide the reader through the complex mechanisms derived from stress research and progressively connect these to the field of psychosis, utilizing the translational potential of preclinical studies. The goal is to help the reader navigate the intricate pathways of medicine while appreciating a holistic approach to understanding the pathobiological underpinnings of psychosis. To achieve this, the review needs to include a substantial amount of preclinical findings that serve as the theoretical foundation and aid the reader in comprehending similar mechanisms in psychosis. To further support this approach, the author prompts Reviewer 1 to examine a notable paper by a Canadian group, Rawani et al. (Antioxidants 2024, 13(6)), which reviews the multisystemic substrate of psychosis and dedicates significant attention to the impact of each system on the central nervous system and neurotransmitters based on preclinical evidence. The Author, taking into account the Reviewer's 1 suggestion to condense the text in the  second sections, revised the manuscript. The goal was to strike a balance between reducing the amount of information and maintaining the scientific integrity of the text. Finally, the author is not aware of any other paper that addresses this field with such a comprehensive approach, considering numerous variables and scopes—from preclinical evidence to psychosis outcomes—and analyzing the mechanisms both in depth and broadly, while critically highlighting the obscurities and gaps in the current literature, which remains in its early stages. In response to Reviewer’s 1 relatively low rating of the manuscript's scientific soundness, the author expresses frustration and confusion, noting that another reviewer rated the same manuscript highly (5/5). The author deeply respects Reviewer’s 1 scientific view, feels confident that Reviewer 1 will appreciate the merit of this review, and once again expresses sincere thanks for their valuable input.

  Using illustration to summarize the key concepts of the review is good but the only figure is in poor quality, both in contents (eg., unclear/confusing concepts) and in illustrative design. 

Reply:

The figure has been replaced by another with more clarity and an improved illustrative design  

Reviewer 2 Report

Comments and Suggestions for Authors

This review is strikingly similar to the review published by Dr. Karanikas. 

The first two sections are not essential and include substantial redundancy.

The findings presented do not substantially advance what is already in the literature; and reviewed in the literature. 

Author Response

Reviewer 2

This review is strikingly similar to the review published by Dr. Karanikas. 

Reply:

The Author would like to first acknowledge and thank Reviewer 2 for the time and effort dedicated to the review process. Regarding the observation that the manuscript under revision bears notable similarities to a review published by Dr. Karanikas, the Author is uncertain about which specific review Reviewer 2 is referring to. The current under review manuscript cites six previous reviews published in reputable journals, all authored by Dr. Karanikas E, highlighting the author's long-standing involvement and expertise in psychosis pathobiology.

Karanikas, E. The immune-stress/endocrine-redox-metabolic nature of psychosis’ etiopathology; focus on the intersystemic pathways interactions. Neurosci. Let. 2023, 794, 137011.

Karanikas, E. Psychologically traumatic oxidative stress; a comprehensive review of redox mechanisms and related inflammatory implications. Psychopharmacol. Bull. 2021, 51(4), 65.

Karanikas, E. The Gordian knot of the immune-redox systems’ interactions in psychosis. Inter. Clin. Psychopharmacol. 2023, 38(5), 285-296.

 Karanikas, E., Daskalakis, N.P., Agorastos, A. Oxidative dysregulation in early life stress and posttraumatic stress disorder: a comprehensive review. Brain Sci. 2021, 11, 723.

Karanikas. E,. Antoniadis. D,. Garyfallos. G.D. The role of cortisol in first episode of psychosis: a systematic review. Curr. Psychiatry Rep2014, 16(11), 503

Karanikas, E. Cortisol and cytokines in early psychosis, do they correlate? A scoping review. Neurol. Psych. Brain Res. 2019, 32, 91-98

 The Author is open to understanding which specific elements within the previous reviews might resemble the current manuscript, ideally in a point-by-point and sentence-by-sentence format. Given that the Author's research interests align historically with the pathobiological foundations of psychosis, it is natural for the structure and related research topics to be similar. This should not be interpreted as self-duplication; rather, the Author aims to expand upon and enrich the existing body of knowledge. The Author recognizes that the figure in the manuscript might raise concerns about duplication relative to a previously published figure in Dr. Karanikas's review in Neuroscience Letters (2023). In light of this, the Author has taken steps and replaced that figure. Additionally, the Author would like to inform Reviewer 2 that the Editor checked the results of the duplication check process. Aside from a few minor, dispersed issues throughout the manuscript, no significant similarities were identified. In total, there were no concerns regarding duplication in terms of expression or content. The Author has addressed these minor issues, ensuring that the manuscript is now free of any problems. Lastly, the Author wishes to convey to Reviewer 2 that, as the Author aims to elaborate, delineate, and deepen the understanding of his scientific field, it is expected that the structure of the manuscript may reflect previous papers. This is part of an ongoing effort to explore and enhance prior knowledge. Once again, the Author expresses gratitude to Reviewer 2 for their contributions to the review process.

The first two sections are not essential and include substantial redundancy.The findings presented do not substantially advance what is already in the literature;  and reviewed in the literature.

Reply: The author wishes to extend gratitude to Reviewer 2 for their valuable input and takes their comments seriously. As outlined in the title of this manuscript under review, "The Labyrinth of the Immune-Redox-Metabolic-Neuroendocrine/Stress Multisystemic Interplay: From Stress to Psychosis Pathobiology," this review aims to evaluate the existing evidence regarding the involvement of these various factors in both stress and psychosis. Based on over 20 years of expertise in the field, the author’s prior impression is that systematic research on the role of each one of the reviewed systems in psychosis has only emerged over the last decade or two. Nonetheless, references detailing how these systems are interconnected in forming the substrate of psychosis remain minimal, if they exist at all. This observation motivated the author to embark on this research journey. Historically, the theoretical framework explaining how these systems function has primarily been based on preclinical studies related to stress. For a long time, stress has been viewed as a critical catalyst impacting brain structures and other bodily organs. The neural diathesis-stress model, conceptualized by Walker and Diforio in 1997, encapsulates the close relationship between stress, brain dysfunction, and the likelihood of experiencing psychosis. Since then, researchers have systematically explored the neurobiological foundations of psychosis, leading to an era in which major fields of medicine—such as metabolism, immunity, redox biology, and neuroendocrinology—are under constant investigation to better understand the pathobiological substrate of psychosis. As knowledge of each proposed pathobiological pillar expands, novel factors, including neurosteroids, gonadal steroids, and novel metabolic factors, are continually added to the discussion surrounding the underpinnings of psychosis. Presently, researchers are on the cusp of an era in which a multisystematic approach to understanding pathobiology is gaining prevalence. Consequently, readers of academic papers are presented with a wealth of information covering various scopes. This situation has encouraged the author to integrate knowledge from multiple areas believed to contribute to the pathobiology of psychosis. Starting with stress as a foundation, the author aims to guide the reader through the complex mechanisms derived from stress research and progressively connect these to the field of psychosis, utilizing the translational potential of preclinical studies. The goal is to help the reader navigate the intricate pathways of medicine while appreciating a holistic approach to understanding the pathobiological underpinnings of psychosis. To achieve this, the review needs to include a substantial amount of preclinical findings that serve as the theoretical foundation and aid the reader in comprehending similar mechanisms in psychosis. To further support this approach, the author prompts Reviewer 2 to examine a notable paper by a Canadian group, Rawani et al. (Antioxidants 2024, 13(6)), which reviews the multisystemic substrate of psychosis and dedicates significant attention to the impact of each system on the central nervous system and neurotransmitters based on preclinical evidence. Finally, the author is not aware of any other paper that addresses this field with such a comprehensive approach, considering numerous variables and scopes—from preclinical evidence to psychosis outcomes—and analyzing the mechanisms both in depth and broadly, while critically highlighting the obscurities and gaps in the current literature, which remains in its early stages. Yet, the Author, taking into account the Reviewer's 2 suggestion to condense the text in the first and second sections, revised the manuscript. The goal was to strike a balance between reducing the amount of information and maintaining the scientific integrity of the text. In response to Reviewer 2's low rating of the manuscript's scientific soundness, the author expresses frustration and confusion, noting that another reviewer rated the same manuscript highly (5/5). The author deeply respects Reviewer’s 2 scientific view, feels confident that Reviewer 2 will appreciate the merit of this review, and once again expresses sincere thanks for their valuable input.

Reviewer 3 Report

Comments and Suggestions for Authors

The authors presented an article titled “The labyrinth of the immune-redox-metabolic-neuroendocrine/stress multisystemic interplay; from stress to psychosis pathobiology” that aims to describe psychosis based on the inflammatory hypothesis etiopathology findings from each one of the immune-redox-metabolic-neuroendocrine/stress systems. 

The authors are encouraged to address the following concerns:

1. Based on your statements that: “Nuclear factor kappa-light-chain-enhancer of activated B cells (NF-κB)” resulting in psychosis, what pharmacological interventions would be best used to treat psychosis beyond the current psychotropics in psychiatry?

2) Based on the NF-κB basis for psychosis, what existing psychotropics inhibit this pathway?

3) Please comment on whether Electroconvulsive Therapy (ECT) inhibits any of the immune-redox-metabolic-neuroendocrine/stress systems.

4) Based on the age of onset of schizophrenia in males versus females, please comment on the possible factors leading to modulation of the immune-redox-metabolic-neuroendocrine/stress system resulting in psychosis?

5) Does marijuana result in damage to the immune-redox-metabolic-neuroendocrine/stress system?

6) Comment on whether any traditional teas/herbs that play a role in mitigating damage to the immune-redox-metabolic-neuroendocrine/stress system.

7) In the sub-heading titled: “Metabolism/Energy bioavailability and CNS” please comment on how diabetes or glucose metabolism or ketone energy use may influence psychosis in Dementia in the elderly. 

8) In the sub-heading titled: “Immune-redox-metabolic-neuroendocrine/stress systems in psychosis-clinical evidence” you mention the following inflammatory cytokines IL-1, IL-6, TNF-a. During the COVID-19 pandemic there was an article that identified the potential use of fluoxetine to treat patients with COVID-19 at a dose of 20 mg per day. Would fluoxetine be a treatment for psychosis?

9) In the sub-section titled: “Lipids metabolism and psychosis” please comment on whether -statin drugs are associated with psychosis or not to support your hypothesis. 

10) Knowing the efficacy of antipsychotics and how so many of the 2nd generation antipsychotics causing weight gain, is this a natural process to heal the brain and decrease inflammation?

Author Response

Reviewer 3

The authors presented an article titled “The labyrinth of the immune-redox-metabolic-neuroendocrine/stress multisystemic interplay; from stress to psychosis pathobiology” that aims to describe psychosis based on the inflammatory hypothesis etiopathology findings from each one of the immune-redox-metabolic-neuroendocrine/stress systems. 

Reply: The Author would like to express gratitude to Reviewer 3 for their time and effort in reviewing the manuscript. Special thanks are extended for the high rating awarded concerning the manuscript’s scientific soundness. Additionally, the Author appreciates Reviewer’s 3 suggestions, which focus on intriguing topics related to the multisystemic aetiopathogenetic hypothesis of psychosis. Each suggestion offers a unique perspective that could lead to separate studies in the future. The Author will certainly keep Reviewer’s 3 suggestions in mind for potential future research. However, due to the word limit of the manuscript, it is not possible to provide an extensive discussion of each suggestion at this time. Lastly, the Author would like to note that the other Reviewers have recommended a reduction in the manuscript’s length.       

The authors are encouraged to address the following concerns:

  1. Based on your statements that: “Nuclear factor kappa-light-chain-enhancer of activated B cells (NF-κB)” resulting in psychosis, what pharmacological interventions would be best used to treat psychosis beyond the current psychotropics in psychiatry?

Reply: it has been added Beyond the conventional limits of psychotropic medications, other agents with immunomodulating properties, such as NSAIDs, COX inhibitors, minocycline, and N-Acetylcysteine (NAC), have shown beneficial effects in early psychosis as adjunct agents [115]

2) Based on the NF-κB basis for psychosis, what existing psychotropics inhibit this pathway?

Reply: it has been addressedClozapine, as well as 2nd generation antipsychotics such as olanzapine and risperidone, and Lithium have exhibited downregulatory potential over NF-kB, yet not consistently [111,112,113]. Thus, the pharmacological field provides evidence of the antipsychotics’ anti-inflammatory effect, hence corroborating the inflammatory etiopathological basis in psychosis.”

3) Please comment on whether Electroconvulsive Therapy (ECT) inhibits any of the immune-redox-metabolic-neuroendocrine/stress systems.

Reply: it has been addressed : “Golden standard methods for refractory states in psychosis, such as Electroconvulsive Therapy (ECT), have only recently started being associated with immunomodulating effects, as evidenced by IL-6 and CRP decrease [116]

4) Based on the age of onset of schizophrenia in males versus females, please comment on the possible factors leading to modulation of the immune-redox-metabolic-neuroendocrine/stress system resulting in psychosis?

Reply: it has been addressed :Having spoken about the possible immune components’ moderating role in the emergence of psychosis, gender, and in particular the male sex, and its association with certain C4 risk haplotypes for schizophrenia, is hypothesized to underlie the higher frequency of men in early onset psychosis [106].”

and “Interestingly, oestrogen and their D2-Rs dowregulatory potential appears to account for the sex difference in age of schizophrenia onset [250]

5) Does marijuana result in damage to the immune-redox-metabolic-neuroendocrine/stress system?

Reply: it was addressed : “ Last but not least, misuse of psychotropic substances like marijuana further hampers any attempt to disentangle the multisystemic directionality, as recent studies are suggestive of the latter’s pro-inflammatory/oxidative/gluco-dysregulatory/hypercortisolemic effects [286, 287,288].”  

6) Comment on whether any traditional teas/herbs that play a role in mitigating damage to the immune-redox-metabolic-neuroendocrine/stress system.

Reply: it was addressed : “Similarly, other recreational substances (Curcumin, green tea, ginseng) have been suggested interfering with the immune-redox-metabolic-neuroendocrine/stress systems, either in an  anti-inflammatory fashion, hence possibly prophylactic against psychosis, whereas others (Ayahuasca, kratom) in an opposite way[289,290,291].”

7) In the sub-heading titled: “Metabolism/Energy bioavailability and CNS” please comment on how diabetes or glucose metabolism or ketone energy use may influence psychosis in Dementia in the elderly. 

Reply: The author would like to thank Reviewer 3 for their thoughtful comment. It is true that glucose dysregulation and the use of ketone energy have increasingly been recognized as significant factors affecting dementia-related psychosis. The relationship between metabolic dysfunction and the pathobiology of psychosis in the elderly is indeed fascinating. However, any references to this intriguing field are considered out of scope for the current manuscript under review. This review aims to focus on the mechanisms related to the pathobiology of psychosis as early as possible in the disorder's emergence, in order to eliminate any heterogeneity related to chronicity, medication, and comorbidity factors. The author will certainly keep Reviewer 3's suggestions in mind for future studies.

8) In the sub-heading titled: “Immune-redox-metabolic-neuroendocrine/stress systems in psychosis-clinical evidence” you mention the following inflammatory cytokines IL-1, IL-6, TNF-a. During the COVID-19 pandemic there was an article that identified the potential use of fluoxetine to treat patients with COVID-19 at a dose of 20 mg per day. Would fluoxetine be a treatment for psychosis?

Reply: it was addressed: Interestingly, even antidepressant medications, through their action on the STAT/JAK pathway, which is thought to participate in the pathobiology of psychosis, may have an adjunctive therapeutic role against psychosis [114]”

Certain domains, particularly the Kynurenine pathway, are involved in regulating intersystemic interactions. Inflammatory cytokines enhance the conversion of tryptophan to kynurenines. Kynurenic acid (KYNA) affects neurotransmitters such as Glutamate, Dopamine, Acetylcholine (Ach), and GABA by antagonizing α7 nicotinic Ach–Rs. Additionally, KYNA blocks NMDA-Rs on PVI, disrupting midbrain dopaminergic signaling [285]. This creates a vicious cycle of inflammation, metabolism (tryptophan to kynurenines), and neurotransmitter imbalance. This underlying pathobiological vicious mechanism may explain the close association of psychosis with depression.” 

9) In the sub-section titled: “Lipids metabolism and psychosis” please comment on whether -statin drugs are associated with psychosis or not to support your hypothesis. 

Reply: It was addressed “This ambiguity surrounding the potential role of lipids in the aetiopathogenesis of psychosis is supported by the conflicting results of two recently published meta-analyses [106,214]. Conversely, there is no substantial evidence to suggest risk for the emergence or aggravation of already diagnosed psychosis in subjects under statins [215].”  

10) Knowing the efficacy of antipsychotics and how so many of the 2nd generation antipsychotics causing weight gain, is this a natural process to heal the brain and decrease inflammation?

Reply: It was addressed:A deeper understanding of the role that various biological parameters play in the underlying pathology of psychosis could advance the field of precision psychiatry. This could lead to the development of new pharmacological agents that specifically target the immune, redox, metabolic, and neuroendocrine/stress systems. In addition, everyday lifestyle practices—such as a diet rich in omega-3 fatty acids, herbs with antioxidant properties, gut health, regular exercise, and mindfulness techniques—are beginning to show their potential benefits for maintaining balance in these systems. [289,290,292,293]. Together, these approaches could enhance the toolkit available for treating psychosis.” 

Round 2

Reviewer 1 Report

Comments and Suggestions for Authors

The revised version has improved significantly.

Reviewer 2 Report

Comments and Suggestions for Authors

Overall concerns remain the same. The first 4 sections have already been extensively reviewed in the literature. 

Section 5 is the only section with potential in terms of presenting novel review material.

Fig. 1 has been rotated, color added, and a reduction in crisscrossed arrows which makes for better presentation. However, the content still seems to be the same as presented in previous publication. Additionally, the directionality of select arrows differs from initial version.  It is unclear if hypothesis has changed or if the arrows were entered incorrectly. For example please see Neuroendocrine - GCs....-HPA, SAM for example.